# Safety and Immunogenicity of a Stable, Cold-Adapted, Temperature-Sensitive/Conditional Lethal Enterovirus A71 in Monkey Study

**DOI:** 10.3390/v13030438

**Published:** 2021-03-09

**Authors:** Kaw Bing Chua, Qimei Ng, Tao Meng, Qiang Jia

**Affiliations:** Temasek Life Sciences Laboratory, 1 Research Link, National University of Singapore, Singapore 117604, Singapore; qimei_85@yahoo.com.sg (Q.N.); mengtao@tll.org.sg (T.M.); jiaqiang@tll.org.sg (Q.J.)

**Keywords:** enterovirus A71, coxsackievirus A16, attenuation, temperature sensitive and conditional lethal, candidate vaccine

## Abstract

Enterovirus A71 (EV-A71) and coxsackievirus A16 (CA16) are major etiological agents of hand foot and mouth disease (HFMD) in children, which may result in fatal neurological complications. The development of safe, cost effective vaccines against HFMD, especially for use in developing countries, is still a top public health priority. We have successfully generated a stable, cold-adapted, temperature sensitive/conditional lethal EV-A71 through adaptive culturing in Vero cells at incrementally lower cultivation temperatures. An additional 40 passages at an incubation temperature of 28 °C, and a temperature reversion study at an incubation temperature of 37 °C and 39.5 °C, reveals the virus’s phenotypic and genetic stability at the predefined culture conditions. Six unique mutations (two in noncoding regions and four in nonstructural protein-coding genes) in combination may have contributed to its stable phenotype and inability to fully revert to its original wild phenotype. The safety and immunogenicity of this stable, cold-adapted, temperature sensitive/conditional lethal EV-A71 was performed in six monkeys. None of the inoculated monkeys developed any obvious clinical illness except one which developed a transient spike of fever. No gross postmortem lesion or abnormal histological finding was noted for all monkeys at autopsy. No virus was reisolated although EV-A71 specific RNA was detected in serum samples collected on both day 4 and day 8 postinoculation. Only EV-A71 RNA and viral antigen were detected in the spleen homogenate and peripheral blood mononuclear cells, respectively, collected on day 4. The two remaining monkeys developed good humoral immune response on day 14 and day 30 post-inoculation.

## 1. Introduction

Hand foot and mouth disease (HFMD) is a febrile illness complex characterized by cutaneous maculo-papulo-vesicular eruptions (exanthem) involving the palms and soles of the hands and feet, with simultaneous occurrence of muco-cutaneous vesiculo-ulcerative lesions (enanthem) affecting the mouth. The illness is caused by several enteroviruses with coxsackievirus A16 (CA16) and enterovirus A71 (EV-A71) as the main causative agents [1]. Besides HFMD, EV-A71 has been associated with a spectrum of other clinical diseases. These diseases include nonspecific febrile illnesses, acute infantile respiratory infections, aseptic meningitis, poliomyelitis-like acute flaccid paralysis, acute encephalitis and encephalomyelitis [1,2,3,4]. Both EV-A71 and CA16 are small non-envelope positive sense RNA viruses classified into *enterovirus A* species in the *Enterovirus* genus of the *Picornaviridae* family. EV-A71 strains are divided into seven distinct genogroups, A−G, based on the nucleotide sequence of its major capsid protein (VP1). Genogroups B and C are further subclassified into subgenogroups B1-5 and C1-5, respectively, by the same phylogenetic analysis of VP1 gene [4,5,6,7].

In recent decades, EV-A71 infections have become a major public health burden and concern throughout the world, especially in the Asia-Pacific region following the occurrence of epidemic and outbreaks of neurovirulent EV-A71. Neurovirulence of EV-A71 first gained prominent global attention in an outbreak of neurological diseases in Bulgaria in 1975, which caused 705 cases of poliomyelitis-like disease with 44 deaths [8,9]. An outbreak of an illness of a similar nature occurred in Hungary in 1978, resulting in many cases of poliomyelitis-like illnesses and 47 deaths [10]. Subsequently, several epidemics of HFMD associated with milder central nervous system (CNS) complications due to EV-A71 were reported in Australia, New York, Hong Kong, and Philadelphia [11,12,13,14,15,16]. In Japan, two epidemics of EV-A71 have occurred with most of the cases characterized by HFMD and a low incidence of CNS disease [17,18,19]. In 1997, a large outbreak of HFMD, due to highly neurovirulent EV-A71, emerged in Malaysia and caused 48 deaths [20,21]. A larger outbreak occurred in Taiwan in 1998, with more than 100,000 cases of HFMD, 405 severe infections and 78 deaths due to acute brainstem encephalomyelitis associated with neurogenic cardiac failure and pulmonary edema [22,23,24,25]. A total of 488,955 HFMD cases with 126 deaths were reported on the mainland of the People’s Republic of China in 2008 [26]. The number of HFMD cases was reported to increase to 1,155,525, with 353 fatalities in the 2009 outbreak [27]. In 2010, the country experienced the largest ever HFMD outbreak, with more than 1.7 million cases, 27,000 patients with severe neurological complications and 905 deaths. In all the three outbreaks, almost all the severe cases with neurological complications and deaths were due to EV-A71 [28]. 

Currently, the molecular determinants of virus virulence and the neuro-pathogenesis of EV-A71 infections are still not fully understood. In addition, no antiviral drug has been approved for the clinical treatment of severe infections and associated neurological complications. For control and prevention, the development of cost-effective vaccines against both CA16 and EV-A71, especially for use in developing countries, is a top public health priority. Various types of vaccines against EV-A71 under investigation and development appear to elicit an immune response in rodents or monkeys [29,30,31,32,33,34,35,36,37]. Though virus-like particle vaccines and subunit peptide vaccines based on the VP1 capsid protein remain viable potential strategies worthy of further study and development, the injectable inactivated and oral attenuated EV7-A71 vaccines remain the most promising candidates. This finding is based on vast past experience of the development and usage of inactivated injectable Salk and live attenuated oral Sabin poliovirus vaccines in the control and near eradication of wild-type poliovirus infections [38,39]. 

In this report, we generated a stable, cold-adapted, temperature-sensitive/conditional lethal EV-A71 strain which did not exhibit neuro-virulence in monkeys following intravenous inoculation but induced good immunogenic response. 

## 2. Materials and Methods

### 2.1. Generation and Characterization of Temperature-Sensitive/Conditional Lethal EV-A71

#### 2.1.1. Cell, Virus and Cold-Adaption Process

All cell lines used in this study were obtained from the American Type Culture Collection (ATCC, Manssas, VA, USA). The cell lines were grown in Dulbecco’s modified Eagle’s medium (DMEM, Gibco, Gaitherburg, MD, USA) supplemented with 10% (*v/v*) fetal calf serum (FCS, i-DNA, Singapore) and 0.22% (*w/v*) sodium bicarbonate (NaHCO_3_, Sigma Aldrich, St. Louis, MO, USA). Vero cells CCL81 (ATCC, Manssas, VA, USA) used for virus propagation, attenuation, titration, and assessment of temperature sensitive phenotype were cultured and maintained at an incubation temperature of 37 °C, unless otherwise stated. Prior to infection, the culture medium of DMEM 10% was replaced with DMEM 1%, and the infected cells were subsequently incubated at the respective experimental temperature in a humidified environment with 5% CO_2_. Ten EV-A71 clinical isolates, isolated from oral secretion, stool, vesical fluid and/or tissue specimens of 10 patients who had presented with hand foot and mouth disease (HFMD) with and without neurological complication in the 1997 HFMD outbreak in Malaysia [40]. These were plaque purified and selected in Vero cells at an incubation temperature of 36 °C as previously described [41]. Three EV-A71 isolates were chosen based on larger size and higher uniformity of plaques. The virus stock designated as the respective parental wild-type strain was prepared following two further passages in Vero cells at 36 °C. All virus stocks and successive further passaged strains at incrementally lower incubation temperatures were stored in a minus 80 °C freezer.

Freshly confluent monolayer Vero cells in DMEM 1% FCS were used to adapt the virus strains for replication at successively incrementally lower temperatures, starting from an initial temperature of 34 °C and going down in steps of 1 °C. The clarified culture supernatant containing the virus was passed into each successive fresh new culture flask of Vero cells as soon as it attained full cytopathic effect (CPE). The culture supernatant containing the virus was passed at a higher multiplicity of infection (m.o.i) of 20 at the beginning of each successive change to a lower temperature of incubation. After it was noted to be able to cause full CPE in Vero cells within 3 days after inoculation, the new culture flask containing freshly confluent monolayer Vero cells was inoculated with the same m.o.i of virus for three more passages before subsequently being reduced to a lower m.o.i of 5. Once it was noted to be able to cause full CPE within 3 days with an inoculum of 5 m.o.i, the attenuation process was moved on to the next successive incremental 1 °C lower incubation temperature after an additional three passages. 

#### 2.1.2. Virus Titration

Virus titre was determined by microtitration assay in Vero cells in accordance with the method described in the 2004 Polio Laboratory Manual from the World Health Organization, with minor modification. Virus titre was calculated as 50% cell culture infectious dose (CCID_50_) per milliliter following the method of Reed and Muench (1938) [42]. Briefly, following treatment with an equal volume of chloroform to disperse virus aggregates, a 10-fold serial dilution of the clarified virus supernatant was made in DMEM 1% FCS. Vero cell monolayers (10^4^ cells per well) in a 96-well flat-bottom tissue culture plate were inoculated with 100 µL of the serially diluted virus stock and incubated in 5% CO_2_ at each respective incubation temperature for 5 days prior to observation for the presence of CPE. The assay was independently repeated 4 times.

#### 2.1.3. Temperature-Sensitivity Phenotype Assay

Two approaches were used to assess the growth characteristics of the virus strains in Vero cells at incubation temperatures of 28, 37 and 39.5 °C. The first approach assessed the number of days taken for the virus strain to cause full CPE in infected cells (cell death kinetic), and the second approach assessed the titre of the virus at full CPE in cells incubated at each specific temperature tested. Briefly, in the first approach, the growth medium of three T-25 tissue culture flasks containing a confluent monolayer Vero cells of similar age were replaced with DMEM 1% FCS. The medium in each flask was allowed to equilibrate to the specified temperature to be tested by placing in respective incubators for 1 hour, prior to inoculation with the virus strain at an inoculum of 10 m.o.i. If no CPE was observed at the end of 10-day’s culture, the supernatant was passed into a new flask of monolayer Vero cells and similarly incubated for another 10 days. It was taken as no virus replication if no CPE was noted following the second passage. In the second approach, a Vero cell suspension of 10^4^ cells per 100 µL was seeded into each well of three 96-well cell-culture plates and incubated at 37 °C, 5% CO_2._ After 10 hours of incubation, each cell-culture plate was then allowed to equilibrate to the specific temperature to be tested by placing in respective incubators for 1 hour. The cells in each well were subsequently inoculated with 100 µL of 10-fold serial dilutions of the virus strain before being transferred to incubators of respective temperature and incubated for 5 days prior to observation for the presence of CPE, virus titre was calculated as the 50% cell culture infectious dose (CCID_50_) per milliliter following the method of Reed and Muench (1938) [42]. 

#### 2.1.4. Virus Growth Kinetics by Total Viral RNA Quantity in the Culture Supernatant

DMEM 10% FCS of freshly confluent monolayer Vero cells in T-25 flasks was removed and replaced with 1 mL DMEM 1% FCS. After allowing an hour of equilibrium at the respective incubation temperatures of 28, 37 and 39.5 °C, the cells were infected with 10 m.o.i of respective virus strains. The infected cells were washed thrice with sterile PBS (prewarmed to the respective temperature of study) after an hour of preadsorption at respective incubation temperature. A portion of 5.5 mL of DMEM 1% FCS was added after the last wash. An aliquot (150 µL) of culture supernatant was withdrawn immediately and subsequent 12 hourly for 5 days following infection. All aliquots of infected culture supernatants were stored at −80 °C for later assay of viral RNA quantity by qRT-PCR. Two independent experiments were performed for each virus strain.

Aliquots of culture supernatants were clarified by centrifugation and viral RNA was extracted and purified by E.Z.N.A. Viral RNA Kit (Omega Bio-tek, Nocross, CA, USA). TaqMan quantitative real-time PCR was performed as described previously [43]. The viral RNA was quantified by QuantiNova Probe RT-PCR kit using primers qRT-PCR-EV-F (5′ AATAAATCATAACCTCCGGCCCCTGAATG 3′) and qRT-PCR-EV-R (5′ AATAAATCATAAGAAACACGGACACCCAAAGTAGTC 3′) and a Taqman probe qRT-PCR-EV-probe (5′[6FAM] TCCGCTGCAGAGTTRCCCGTTACGA [TAM]3′). The RT-PCR thermal cycling conditions were applied at an initial incubation at 45 °C for 15 min (reverse transcription), 95 °C for 5 min (initial PCR activation step), followed by 40 cycles: 95 °C for 5 s (denaturation), 60 °C for 30 s (combined annealing and extension) and signal collection. The reaction was carried out using a Rotor-Gene Q real time PCR cycler (Qiagen, Hilden, Germany). An assay of viral RNA quantity was performed in duplicate for culture supernatant and in triplicate for monkey serum samples. The virus load was measured in copy number of RNA which was calculated based on performed standard curve and the error bar was calculated in Microsoft Excel basing on the standard deviation of 4 parallel repeats.

#### 2.1.5. Genetic Stability and Temperature Sensitivity Reversion Assay

To assess the genetic stability of the virus strains cultured in the specified cell-type and environment, the virus strains were further passaged twenty times in Vero cell culture at an inoculum of 5 m.o.i and an incubation temperature of 28 °C. At the end of twenty passages, the virus strains were assessed for their temperature sensitivity by culturing at incubation temperatures of 28, 37 and 39.5 °C as described above. The complete nucleotide sequences of their respective genomes were subsequently sequenced and analyzed with respect to those of their respective parental wild-type viruses.

A temperature sensitivity reversion assay was carried out on a stable, cold-adapted, temperature-sensitive virus strain. The selected strain was passaged 6 times in monolayer Vero cells incubated at 37 °C. At each passage, an inoculum of 10 m.o.i was used. The derived virus strain at each passage was assessed for its growth characteristics in Vero cells at incubation temperature of 28, 37 and 39.5 °C, as described above for temperature sensitivity assay. The complete genomes of virus strains at each successive passage at 37 °C were sequenced and analyzed.

#### 2.1.6. RNA Extraction, RT-PCR and Sequencing

Viral genomic RNA was extracted from the culture fluid of infected cells at full CPE using a commercially available Viral RNA Extraction Kit (Qiagen, Hilden, Germany). First-strand synthesis was performed with EV-A71-specific primers using Superscript II RNA polymerase (Invitrogen, Calrsbad, CA, USA), and subsequent PCR with 18 degenerate primer pairs was performed using GoTaq Green PCR mix (Promega, Madison, WI, USA) (information on primers’ sequences will be available upon request). Fragments generated were sequenced using BigDye Terminator sequencing kit (Applied Biosystems, USA). The 5’Rapid Amplification of cDNA end (RACE) was performed to determine the 5′-UTR viral sequence by ligating the 5′-cordycepin-blocked adaptor DT88 to the 5′-end of the EV71 cDNA using standard T4 DNA Ligase (Fermentas, Vilnius, Lithuania), and was followed by standard PCR using a DT88-complementary primer and an EV71-specific primer. The 3′-RACE was also performed to determine the 3′-UTR viral sequence using an oligo-dT primer [44].

#### 2.1.7. Molecular Cloning and Sequencing of Ambiguous Sequences

Fragments with ambiguous sequences were cloned into pZero-2 plasmid (Invitrogen, Calrsbad, CA, USA) and transformed into TOP10 *E. coli* cells (Invitrogen, Calrsbad, CA, USA). Plasmids containing the cloned EV-A71 fragments were extracted from at least 10 colonies from each transformant using commercially available Plasmid Miniprep Kit (Qiagen, Hilden, Germany) and the sequence of the cloned fragment was subsequently determined. 

Fragment sequences obtained were merged using the European Molecular Biology Open Software Suite (EMBOSS; http://mobyle.pasteur.fr/cgi-bin/portal.py?#forms::merger, accessed on 21 February 2021) [45]. Merged sequences were aligned with a reference sequence of EV-A71 strain 3799-SIN-98 (NCBI accession no. DQ341354.1) using the BioEdit Sequence Alignment Editor v. 7.0.9.0 [46].

### 2.2. Monkey Study

The monkey study on safety and immunogenicity of the selected cold-adapted, temperature-sensitive/conditional lethal strain of EV-A71 (EV71:TLLβP20) was contracted to investigators based in the Animal Facility of DUKE-NUS, Singapore. Seven cynomolgus monkeys (*Macaca fascicularis*) free of *Mycobacterium tuberculosis* and simian immunodeficiency virus, three females (2202F, 2207F, 2891F) and four males (0791M, 2247M, 2889M, 2890M) with a mean weight of 3.23 kg (range 2.44 to 4.11, SD = 0.7) were used to study the safety and immunogenicity of the stable, cold-adapted, temperature-sensitive/conditional lethal EV71: TLLβP20. All seven monkeys were prescreened for the absence of binding (indirect immunofluorescence) and neutralizing antibodies against EV-A71. The study, and all animal procedures were approved by the Committees for Biosafety, Animal Handling and Ethics of DUKE-NUS, Singapore (SHS-IBC-201, 11 November 2011). Virus inoculation and observation, animal care and necropsy were performed in accordance with the guidelines of the committees.

The necropsies were performed under the supervision of Dr. Ralph Bunte, a board-certified veterinary pathologist. He also evaluated the histology sections in a blinded fashion. Tissue samples were fixed in 4% paraformaldehyde in phosphate buffered saline (PBS) immediately after collection and stored at 4 °C until processing. The tissue embedding, sectioning, and H&E staining was performed by the Biopolis Shared Facilities Histopathology Unit. Unstained slides were also generated and are available for immunohistochemistry staining if it is needed later. Under light anesthesia with ketamine, 1 mL of virus inoculum was intravenously inoculated into the right saphenous vein. Three monkeys (2889M, 0791M and 2891F) were given an intravenous dose of EV71:TLLβP20 at 10^7^ CCID_50_ per monkey; another three (2890M, 2202F and 2247M) were given 10^8^ CCID_50_ each and the seventh monkey served as negative control. The monkeys were observed twice daily for clinical illness and their body temperature was recorded by an implanted temperature sensor. The stool from each monkey was collected daily and stored at −80 °C for virus isolation later. Two monkeys, one from each viral inoculum dose, were sacrificed under deep anesthesia at day 4 postinoculation (PI). During the autopsy, various parts of central nervous system (CNS), non-neural tissues and blood were collected for histopathological and virological analysis. On day 8 PI, another similar set of two monkeys were sacrificed and the same types of tissues and blood were collected. The remaining two monkeys were given the respective equivalent booster dose of EV71:TLLβP20 at day 14 PI after blood samples were collected for assessment of anti-EV-A71 antibodies. They were then euthanized 16 days after receiving the booster dose and CNS tissues were collected for histopathological study at necropsy.

#### 2.2.1. Histology and Immunohistochemistry

CNS tissue specimens (cerebrum, cerebellum, basal ganglia, brain-stem and spinal cord) and non-neural tissues (lymph node, spleen, liver, kidney, lung and heart) were fixed in 4% paraformaldehyde in phosphate buffered saline (PBS) and embedded in paraffin after fixation. The spinal cord was sectioned horizontally 10 times at cervical, 8 times at thoracic and 10 times at lumbar levels respectively. Paraffin sections, 6 µm thick, were stained with hematoxylin and eosin (H&E) and with Luxol-fast blue/cresyl violet (Kluver-Barrera method) after paraffin removal and rehydration. 

#### 2.2.2. Antigen Detection and Virus Isolation from Monkey Specimens

Blood specimens were collected in both plain and heparinized tubes. Peripheral blood mononuclear cells (PBMC) were harvested from heparinized blood using Ficoll-Paque PLUS (GE Healthcare, Wauwatosa, USA). After two washes with sterile PBS, aliquots of PBMC suspension were seeded onto wells of a Teflon-coated slide for detection of EV-A71 antigen by indirect immunofluorescent assay using a commercial monoclonal antibody (Cat. No. 3324, Sigma Aldrich, St. Louis, MO, USA). Virus isolation was carried out by inoculating PBMC suspension into the wells of 24-well cell culture plate containing monolayer Vero cells. Virus isolation from each serum specimen was performed by inoculating 50 and 100 µL of serum into wells of a 24-well culture plate containing monolayer Vero cells. 

The monkeys’ tissues were lightly washed with two exchanges of sterile PBS and homogenized by grinding using a mortar and pestle inside a Class II biosafety cabinet. Tissue homogenates (10%, *w/v*) prepared in DMEM 1% was clarified by centrifugation at 1000× *g* for 10 minutes. The clarified supernatant was filtered through a 0.22 µm syringe filter, and virus isolation was performed by inoculating 100 and 200 µL of the filtrate. 

For stool samples, a 10% suspension was made in PBS and clarified by centrifugation at 1000× *g* for 10 minutes. After filtration through a 0.22 µm syringe filter, virus isolation was performed by inoculating 100 and 200 µL of stool filtrate. All virus isolation work on monkeys’ samples was performed in duplicate with one set of the inoculated cell culture incubated at 28 °C and the other at 37 °C.

#### 2.2.3. Molecular Detection and Complete Genome Sequencing of Monkey Specimens

A commercial viral RNA extraction and purification kit (Qiagen, Hilden, Germany) was used to extract viral genomic RNA from serum, PBMCs and clarified tissue homogenates. A commercial one-step RT-PCR kit (Qiagen, Hilden, Germany) and a consensus oligonucleotide primer-pair (Sense: 5′-CACCCTTGTGATACCATGGATCAG-3′, Anti-sense: 5′-GTGAATTAAGAACRCAYCGTGTYT-3′) that amplified the proximal third of the VP1 gene from all EV-A71 subgenogroups were used for molecular amplification and detection of EV-A71-specific viral RNA after extraction from tissues. Eighteen pairs of sequence-specific primers, based on the complete genomic sequence of EV71:TLLβ, were used to amplify and sequence the complete genome of EV-A71 present in the sera of two monkeys obtained on day 4 and day 8 PI. Any PCR-amplified fragment that failed to give a good sequence read by direct sequencing was cloned into pZero-2 (Invitrogen, Calrsbad, CA, USA) and transformed into TOP10 *E. coli*. At least ten colonies were selected from each transformant and sequencing was performed on the purified plasmids carrying the inserts. 

#### 2.2.4. Serum Binding and Neutralizing Antibodies Assay

Vero cells infected with EV-A71 subgenogroup B3 were harvested at near full CPE and washed five times with sterile PBS. After the last wash, the suspension of infected cells was mixed with washed noninfected Vero cells in the ratio of approximately 4 infected cells to one noninfected cell. Ten microliters of the mixed cell suspension, containing 250 cells, was carefully layered onto each well of the 12-well Teflon coated slide and allowed to dry over a warm plate. The dried slide was fixed for 10 minutes in cold acetone and used as an inhouse antigen assay for EV-A71 antibodies, starting from an initial dilution of 1:10 for IgM and 1:20 for IgG, by indirect immunofluorescent assay. In the assay for anti-EV-A71 IgM titre, the monkeys’ sera were treated with an appropriate concentration of protein A (Invitrogen, Calrsbad, CA, USA) to remove IgG prior to carrying out serial 2-fold dilution with sterile PBS.

The neutralizing antibody titre of the monkeys was determined by a microneutralizing assay using Vero cells in accordance with the method described in the 2004 Polio Laboratory Manual of World Health Organization with minor modification. The concentration of each EV-A71 subgenogroup used for neutralization was 100 CCID_50_ per 100 µL. The virus neutralization assay was performed using 96-well flat bottom culture plates. A serial 2-fold dilution of each serum sample was prepared in duplicate at a volume of 100 μL DMEM 1% FCS, starting at a 1:10 dilution. An equal volume (100 μL) of virus working stock in DMEM (1% FCS) containing 100 CCID^50^ EV-A71 was added into each well of serially diluted sera and incubated for two hours at 37 °C. After incubation, 100 μL of Vero cell suspension in DMEM 10% FCS, containing 250 cells, was added into each well. The 96-well culture plate was carefully sealed and incubated at 37 °C in 5% CO_2_. The plate was read daily for up to 8 days for presence of CPE. The titre of neutralizing antibodies of each serum sample was determined by the well with the highest dilution that did not show CPE.

## 3. Results

### 3.1. Virus Phenotypic and Genotypic Characteristics in Cells

The wild-type strain of EV71:TLL (subgenogroup B3) was isolated from oral secretion of a child having HFMD. The wild-type strain of EV71:TLLα (subgenogroup B4) was isolated from brain-stem tissue of a child having HFMD associated with fatal acute encephalomyelitis and the wild-type strain of EV71:TLLβ (subgenogroup B3) was from stool specimen of a child having HFMD. Three EV-A71 strains were plaque purified and selected, in Vero cells incubated at 36 °C, from original 10 clinical isolates based on larger plaque size formation and higher uniformity, for the attenuation process. The three selected strains caused full cytopathic effect (CPE) within 3 days in Vero cells incubated at 37 °C at an inoculum of 10 m.o.i. At the same virus inoculum, the virus strains caused full CPE within 5 days in Vero cells incubated at 39.5 °C. All the three strains did not cause CPE in Vero cells incubated at 28 °C, and no CPE was also noted after a blind passage at the end of 10 days culture at 28 °C. The absence of virus replication in inoculated Vero cells at 28 °C was further supported by negative indirect immunofluorescence assay (IFA) of suspended cells harvested from the culture supernatant. 

After more than 90 successive passages at incrementally lower incubation temperatures, all three EV-A71 strains were able to cause full CPE within 3 days in Vero cells incubated at 28 °C at a virus inoculum of ≤5 m.o.i. The virus strains were passaged further at 28 °C until the 100th passage. These virus strains at the 100th passage, derived respectively from the original EV-A71 isolated from oral fluid, brain-stem tissue and stool specimens of 3 different patients, were designated as EV71:TLL, EV71:TLLα and EV71:TLLβ. In a temperature sensitivity assay, all three cold-adapted strains caused full CPE within 2 days in Vero cells incubated at 28 °C at a virus inoculum of 10 m.o.i, but took 4 days to achieve full CPE at an incubation temperature of 37 °C. At an incubation temperature of 39.5 °C, no CPE was noted for all three strains after 10 days of culture. However, EV71:TLL gave 2+ CPE after a blind passage (2nd passage) into a fresh flask of Vero cells that were incubated at 39.5 °C. No CPE was noted for EV71:TLLα and EV71:TLLβ even after two successive blind passages into fresh flasks of Vero cells incubated at 39.5 °C. No suspended cells harvested from culture supernatant of either EV71:TLLα or EV71:TLLβ, inclusive of those derived from two blind passages, gave positive staining with EV-A71 detecting monoclonal antibody. Approximately 1% of floating cells harvested from culture supernatant of EV71:TLL gave positive IFA, though no CPE was noted. 

Incubation temperatures of 28 and 37 °C were used to assay the virus titre of three cold-adapted strains. EV71:TLL gave a virus titre of 1 × 10^8^ CCID_50_/mL and 2 to 3 × 10^7^ CCID_50_/mL at culture temperatures of 28 and 37 °C, respectively. EV71:TLLα gave a virus titre of 1 × 10^8^ CCID_50_/mL at an incubation temperature of 28 °C and 1 to 2 × 10^5.5-6^ CCID_50_/mL at 37 °C. EV71:TLLβ gave a virus titre of 2 to 5 × 10^8^ CCID_50_/mL at an incubation temperature of 28 °C and 1 × 10^7^ CCID_50_/mL at 37 °C.

The stability of cold-adaptation of EV71:TLLα and EV71:TLLβ was assessed by passaging both virus strains for another 20 passages in Vero cells at 28 °C, at an inoculum of 10 m.o.i. After 20 additional passages, EV71:TLLαP20 caused full CPE in cells incubated at 28 °C within 2 days PI, but failed to cause CPE in cells incubated at 37 °C even after 2 blind passages. It retained the ability to achieve a virus titre of 1 × 10^8^ CCID_50_/mL at an incubation temperature of 28 °C. EV71:TLLβP20 maintained the same phenotype after 20 additional passages, at both 28 and 37 °C. The stability of cold-adapted EV71:TLLβ was further assessed by an additional 20 passages (40 additional passages from the 100th passage) under the same culture conditions; and was found to retain the same phenotype.

The virus growth kinetics of EV71:TLLβP20 (Figure 1a) in comparison with those of its parental wild-type ST strain (Figure 1b) by total viral RNA quantity in the culture supernatant of infected Vero cells incubated at 28, 37 and 39.5 °C are shown in Figure 1. The results showed that the virus RNA copy number of EV71:TLLβP20 was consistently highest at incubation temperature of 28 °C and no increment of RNA copy number in the culture supernatant at 39.5 °C from the 12th hour time-point onward. The result corroborated with earlier findings of virus growth kinetics by cell death leading to full CPE at respective culture temperature.

The complete genomes of EV71:TLLα and EV71:TLLβ, their respective strains after an additional 20 passages (EV71:TLLαP20 and EV71:TLLβP20) and original parental wild-types (EV-A71 BS and ST strains) were sequenced and analyzed. The complete nucleotide sequences of the strains were deposited in NCBI, GenBank under respective Accession numbers: MT241233, MT241235, MT241234, MT241236, KF514878 and MT188611. The number of nucleotide changes at each segment of their genes, between EV71:TLLα, EV71:TLLαP20 and original parental wild-type is shown in Table 1. The number of nucleotide changes between original parental wild-type, EV71:TLLβ, EV71:TLLβP20 and EV71:TLLβP40 is shown in Table 2. The detailed number and position of nucleotide and amino acid changes between the original wild-type (ST strain), EV71:TLLβ, EV71:TLLβP20, inclusive of the 5th passage of EV71:TLLβP20 cultured at 37 °C in temperature reversion study, are shown in Appendix A. 

The assessment of reversion from cold-adapted temperature-sensitive phenotype was performed on EV71:TLLβP20 by 6 successive passages in Vero cells incubated at 37 °C. The growth characteristics of the virus derived from each successive culture at 37°C, in terms of virus growth kinetics by cell death (Figure 2a) and virus titre at full CPE (Figure 2b) at incubation temperature of 28 and 37 °C, are shown in Figure 2. The culture supernatant derived from each successive passage for the first 3 passages remained unable to cause any cell death in Vero cells incubated at 39.5 °C. The supernatant derived from 4th to 6th passages caused some cell death, but failed to reach full CPE, and none was able to cause cell death on repassaging into fresh Vero cells incubated at 39.5 °C. The complete genomes of the virus strains derived at each successive passage were sequenced. The number of nucleotide changes/reversions with respect to that of EV71:TLLβP20, are shown in Appendix A.

### 3.2. Monkey Study

#### 3.2.1. Clinical Findings

General observations of clinical status and body temperature of the studied monkeys were performed twice daily. Throughout, all the monkeys were noted to be active and feeding normally. None of the monkeys developed any subtle focal neurological deficit such as limb weakness, tremor or abnormal movement. Only one monkey (2247M) that was given an intravenous dose of 1 × 10^8^ CCID_50_/mL EV71:TLLβP20, developed a spike of fever (39.3 °C) on day 3 postinoculation. No monkeys had weight loss on reweighing at the time they were sacrificed under deep anesthesia.

#### 3.2.2. Autopsy and Histological Findings

No gross postmortem lesion was noted for any monkeys at autopsy. No abnormal histological findings were present in any monkey tissues collected for histopathology.

#### 3.2.3. Virological Investigations

Virus isolation was performed on monkeys’ sera, peripheral blood mononuclear cells (PBMC), stool samples and all autopsied tissues using Vero cells incubated at 28 °C and 37 °C. No virus was isolated from any of the monkeys’ samples despite a blind passage after 10 days culture. One additional blind passage in Vero cells was performed for sera, PBMC samples and those tissue homogenates that were tested positive for EV-A71 by RT-PCR. Despite failing to reisolate the virus, the EV-A71 antigen was detected by IFA in a few PBMCs derived from heparinized blood of two monkeys, 2891F (received 1 × 10^7^ CCID_50_ of EV71:TLLβP20) and 2890M (received 1 × 10^8^ CCID_50_ of EV71:TLLβP20), collected on day 4 PI (Figure 3a). No virus antigen was detected in PBMCs harvested from heparinized blood of monkeys collected on day 8 PI.

EV-A71 specific RNA was detected by RT-PCR in serum samples of all monkeys collected on both day 4 and day 8 PI. Of the non-neuronal tissues, EV-A71 RNA was only detected in the spleen homogenate of 2 monkeys (2202F and 2891F) (Figure 3b). No EV-A71 RNA was detected in any of the neuronal tissue homogenates. The total viral RNA quantity in the serum samples of monkeys collected on day 4, day 8 and day 14 was assayed by quantitative RT-PCR and the result is shown in Figure 4. No viral RNA was detected in the serum samples collected on day 14 from two remaining monkeys (2889M, 2890M). The genome of EV71:TLLβP20, present in the serum samples of two monkeys (2889M and 2890M) collected on both day 4 and day 8 PI, was extracted and completely sequenced. The number of nucleotide and amino acid changes/reversions that occurred in each genomic segment was compared with that of EV71:TLLβP20 (Appendix A).

#### 3.2.4. Monkeys’ Humoral Immune Response

The presence of binding antibodies (IgM and IgG) in the sera of monkeys given intravenous EV71:TLLβP20 was performed by IFA using an in-house prepared antigen. The titres of anti-EV-A71 IgM and IgG present in the blood of the two remaining monkeys (2889M and 2890M) collected on day 14 PI and day 30 PI (16 days post-booster) are shown in Table 3. 

The neutralizing antibodies titre in the serum samples of the two monkeys (2889M and 2890M), collected on day 14 and day 30 PI, against EV-A71 subgenogroups A (BrCr), B3, B4, B5, C1 and C5 was determined by microneutralization assay. The respective titre of neutralizing antibodies of the monkeys’ sera against each of the EV-A71 subgenogroups is shown in Table 4. 

## 4. Discussion

The development of safe, cost effective vaccines against HFMD due to both EV-A71 and CA16, especially for use in developing countries, is still a top public health priority. Three types of formalin inactivated whole virus EV-A71 vaccines have been commercially available in mainland China since 2016 but are not approved for global use [47]. However, oral attenuated vaccines remain promising candidates. This is based on vast knowledge and experience of development and usage of inactivated injectable Salk and live attenuated oral Sabin poliovirus vaccines in the control and near eradication of wild-type poliovirus infections [38,39]. As with live attenuated oral poliovirus vaccines, any live attenuated enteroviral vaccine has many advantages over the inactivated vaccine. These include: cost, avoidance of parental injection and use of adjuvant with its associated side-effects, induction of gut immunity besides induction of both cellular and humoral immunity, induction of more “robust/quality” neutralizing humoral antibodies, and feasibility for mass vaccination within a short time frame. Undeniably, the risk of virus reversion to its wild virulent type is real and of great concern for live attenuated vaccines; however, vast scientific knowledge on the attenuation and virulence reversion of the oral Sabin poliovirus vaccine can be utilized to improve safety in the development of future live attenuated vaccines against other enterovirus infections. 

Although the molecular determinants of virulence and pathogenesis of EV-A71 infections are still not fully understood; the temperature sensitivity phenotype of EV-A71, as with live attenuated Sabin polioviruses, is well known from monkey studies as an important marker of virus neuro-attenuation [38,39,48,49,50,51,52]. Along this line, we generated three strains of EV-A71 which are highly temperature-sensitive, to the extent that they are temperature conditional lethal: thus, providing us with neuro-attenuated EV-A71 strains. The results showed that all three cold-adapted virus strains (EV71:TLL, EV71:TLLα and EV71:TLLβ) caused full CPE in Vero cells within 2 days of incubation at 28 °C and 4 days at 37 °C but failed to induce cell death at 39.5 °C. In virus titre assays, all three virus strains gave a titre of 1 × 10^8^ CCID_50_/mL or higher when cultured at 28 °C. Virus titres were at least 1 log lower when cultured at 37 °C and no measurable virus titre was obtained at 39.5 °C. 

An attenuated poliovirus 1, derived from an oral Sabin poliovirus vaccine (GlaxoSmithKline, London, UK; A0PVB326BC), was propagated in Vero cells at 30 °C and subsequently used in a comparative study of temperature sensitivity [48]. As with Sabin poliovirus 1, EV71:TLL cultured in Vero cells at 39.5 °C showed evidence of breakthrough infection (2+ CPE at 10th day of culture) after a blind passage at the same temperature of incubation. The findings suggest the quasi-species population of EV71:TLL most probably existed as an expanded “vast cloud” with an outlying small subpopulation of virus variants with the ability to rapidly revert to the virulent phenotype [53,54]. Thus, EV71:TLL was not selected for further virus study in the development of this potential live attenuated vaccine candidate, due to concern of rapid reversion to its virulent phenotype. After an additional 20 passages at 28 °C, EV71:TLLαP20 caused full CPE in cells at 28 °C within 2 days PI; but failed to cause CPE in cells at 37 °C, even after 2 blind passages. This phenotypic characteristic is very similar to the reported cold-adapted, highly temperature sensitive attenuated polioviruses by Sanders et al. [55]. EV71:TLLαP20 still retained the ability to achieve a virus titre of 1 × 10^8^ CCID_50_/mL at an incubation temperature of 28 °C. The findings suggest EV71:TLLα remained phenotypically unstable, even in the predefined culture conditions (cell type and incubation temperature). This phenotypic characteristic was confirmed by comparative whole genome sequencing and analysis of its parental wild-type (BS) strain, EV71:TLLα and EV71:TLLαP20, which showed continually higher numbers of nucleotide and amino acid changes in their respective genomes (Table 1). Hence, no further experimental investigation was performed on EV71:TLLαP20 for live attenuated vaccine development, due to its failure to replicate at human body temperature and higher genomic instability. EV71:TLLβ remain phenotypically and genetically stable even following an additional 40 passages at predefined culture conditions (Figure 1, Table 2). 

EV71:TLLβP20 was further assayed for its phenotypic and genetic stability and potential reversion to wild-type, by six successive passages in Vero cells at 37 °C. EV71:TLLβP20 failed to fully revert to its parental wild-type (EV-A71, ST strain) (Figure 2), and was relatively genetically stable; as demonstrated by only one or two nucleotide/amino acid changes from the 3rd passage onward (Appendix A). The significance and molecular basis of the nucleotide change at the 5′ non-coding Cloverleaf, and amino acid reversion in the 2A protein that conferred EV71:TLLβP20 minor degrees of phenotypic reversion, need further investigation to understand the molecular determinants of EV-A71 temperature sensitivity and virulence. 

The genome sequences of the complete genomes of EV71:TLLβ, EV71:TLLβP20 and EV71:TLLβP40 were compared with the reference sequences of their parental wild-type, and with complete genomes of other EV-A71 strains deposited in the GenBank, NCBI. Though EV71:TLLβ had more nucleotide and amino acid changes than EV71:TLLα, in comparison to their respective parental wild-type, the mutations remained stable even after 40 additional passages (Table 1 and Table 2). In comparison to the genomic sequences of other known EV-A71 strains, EV71:TLLβ contained 6 unique mutations that were highly conserved in all isolates. Two of these were in noncoding regions, one in the IRES (T591C) and one in 3′ end (A7400T) of the genome; and four mutations were in nonstructural genes: 2A (C3423T, L31S), 3A (G5262A, A66T) and 3D (G6155T, E72D and C7273T, A445V). The T591C mutation, in the 8-base consensus sequence (590-C***C***TATGGT-597) of the IRES within the domain G (VI), was also highly conserved in polioviruses (consensus octomer 583-C***T***TATGGT-590). This highly conserved 8-base consensus element (CTTATGGT), together with the equally well conserved polypyrimidine tract of the IRES was found to be crucial for translation of poliovirus 1 proteins. In Sabin vaccine poliovirus 1, mutation of the IRES domain V (A480G) and mutation of the 3D viral polymerase coding sequence (U6203G), leading to an amino acid change (H70F), contributed temperature-sensitive attenuation of the virus. One of the six unique nucleotide changes of EV71:TLLβ was in the equivalent genomic region of the 3D polymerase gene (G6155T), and resulted in amino acid substitution from glutamic acid (E) to aspartic acid (D) (E72D) [53,54,56]. However, the molecular basis of how this and other unique or conserved mutations contribute to the observed temperature-sensitive/conditional lethal phenotype of EV71:TLLβ will need further investigation. 

The safety and immunogenicity of EV71:TLLβP20 was tested in a small monkey study. None (except one which showed a transient spike of fever) of the inoculated monkeys developed obvious illness, but all produced a good humoral immune response. Despite the complete virus genomic sequences being retrieved from blood specimens taken on day 4 and 8 PI, and viral antigens being detected in PBMCs by IFA taken on day 4 PI, no virus could be re-cultured in Vero cells incubated at both 28 and 37 °C from these specimens. We speculate that the virus quasi-species population that was so adapted to the defined culture conditions, collapses upon subjection to a new growth environment [57,58]. This may have contributed to our failure to re-isolate EV71:TLLβP20. The intravenous route adopted in this monkey study was based on previous published studies of using this method in assessing the neurovirulence of EV-A71 [49,52]. The shortcoming of this monkey study was the failure to concurrently include the parental wild-type strain of EV71:TLLβP20 for comparison due to financial and ethical constraints. The limitation of this monkey study by intravenous route is that it will not be reflective of EV71:TLLβP20 replication in monkey or human intestinal mucosa.

In summary, we have generated a cold-adapted and highly temperature sensitive, to the extent of being conditional temperature lethal, EV-A71 through adaptive culturing in Vero cells at incrementally lower cultivation temperatures. This EV-A71 strain showed high genetic and phenotypic stability at predetermined culture conditions. A small and limited monkey study demonstrates its neuro-attenuation and immunogenicity. This stable, cold-adapted, temperature sensitive/conditional lethal EV-A71 has the potential for further development into a stable, cold-adapted, temperature sensitive/conditional lethal enteroviral vector and candidate for oral live attenuated vaccine to protect children against EV-A71 infections and HFMD outbreak.

## Figures and Tables

**Figure 1 viruses-13-00438-f001:**
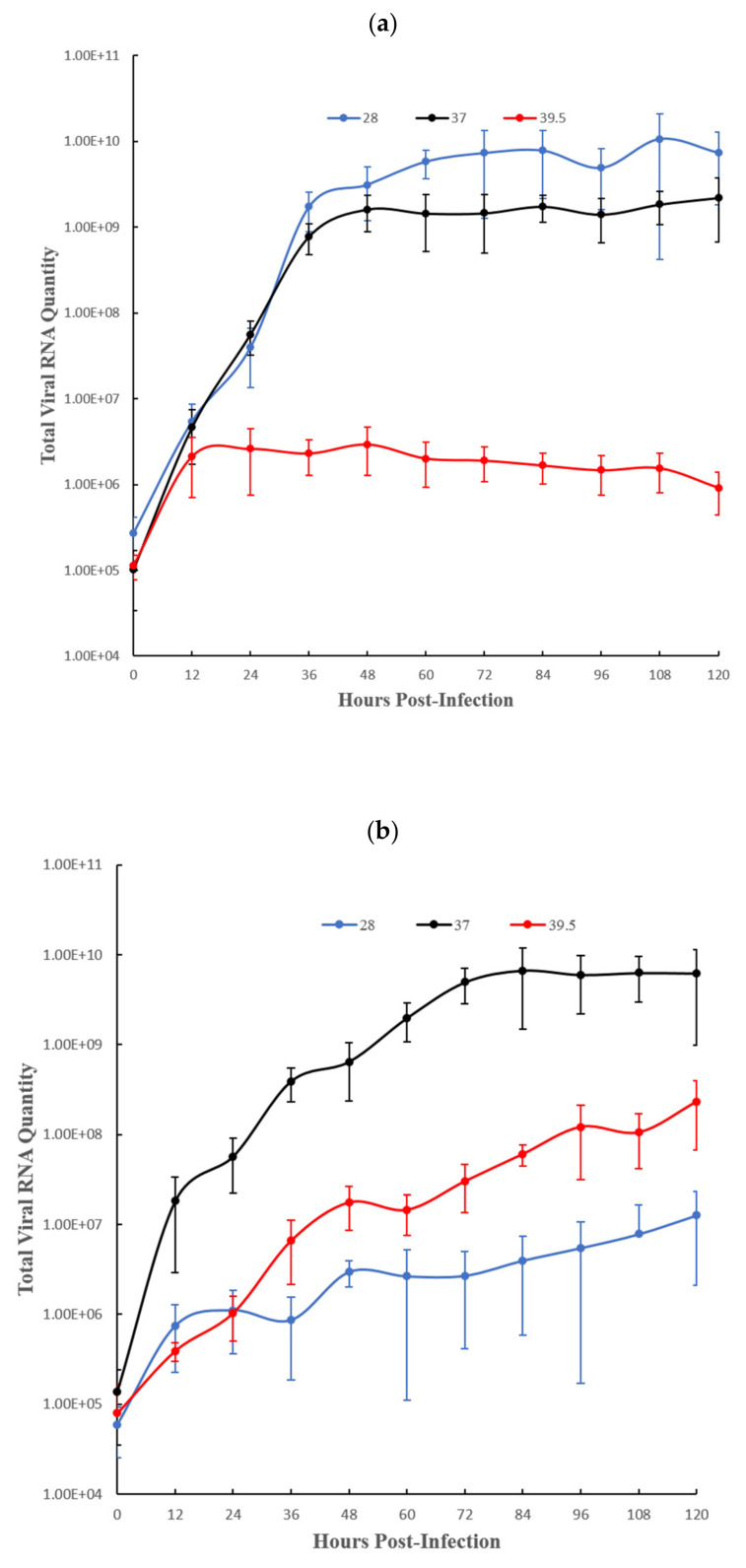
Virus growth kinetics of (**a**) EV71:TLLβP20 and (**b**) its parental wild-type ST strain by total viral RNA quantity in the culture supernatant of infected Vero cells incubated at 28 °C (blue line), 37 °C (black line) and 39.5 °C (red line).

**Figure 2 viruses-13-00438-f002:**
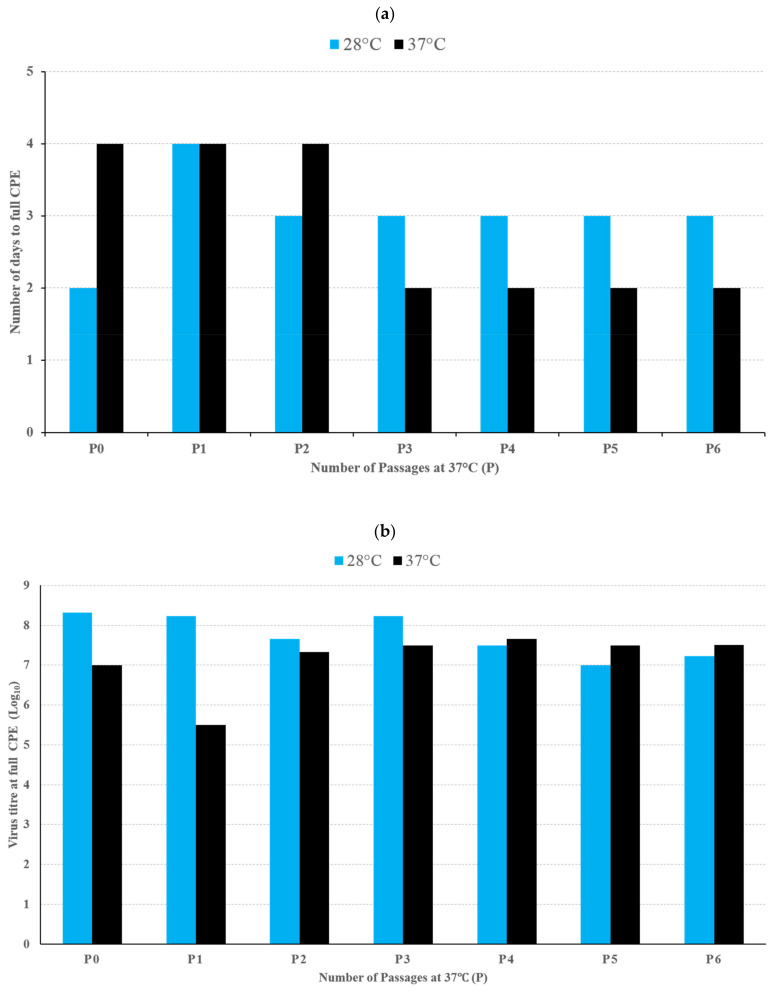
Assessment of temperature reversion growth characteristics of EV71:TLLβP20 following 6 successive passages in Vero cells incubated at 37 °C. Growth characteristics of the virus derived from each successive culture at 37 °C in terms of (**a**) virus growth kinetics by cell death and (**b**) virus titre in CCID_50_/mL at full cytopathic effect (CPE) at incubation temperature of 28 °C (blue bar) and 37 °C (black bar).

**Figure 3 viruses-13-00438-f003:**
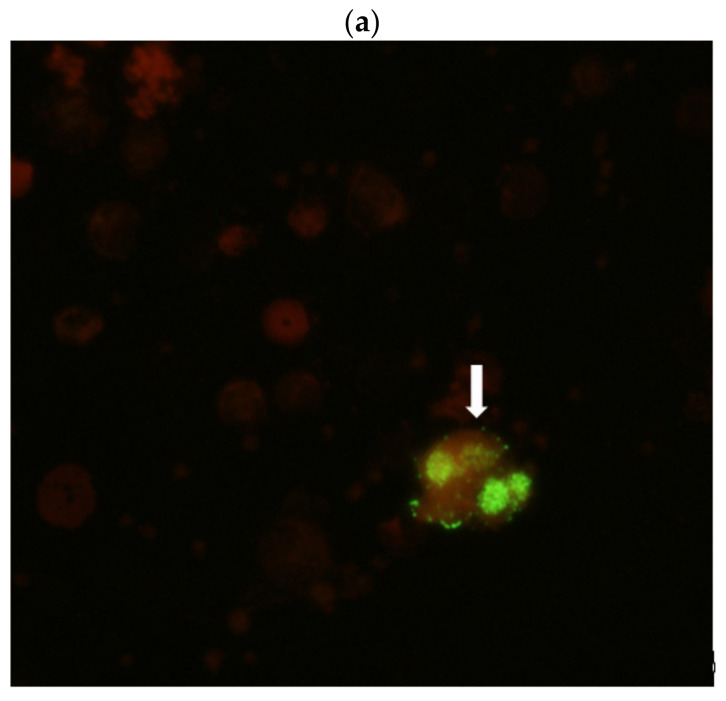
(**a**) A peripheral blood mononuclear cell (arrow), derived from the blood of a monkey on day 4 after being given an intravenous dose of enterovirus A71 (EV71:TLLβP20). Staining for the virus was carried out by indirect immunofluorescence using a commercial monoclonal antibody specific for the virus. (**b**) Photograph of GelRed-stained, electrophoresed agarose gels showing One-step RT-PCR amplified products from tissues derived from day 4 postinoculation monkeys (2202F, 2891F). A specific oligonucleotide primer pair was used for detection of enterovirus A71.

**Figure 4 viruses-13-00438-f004:**
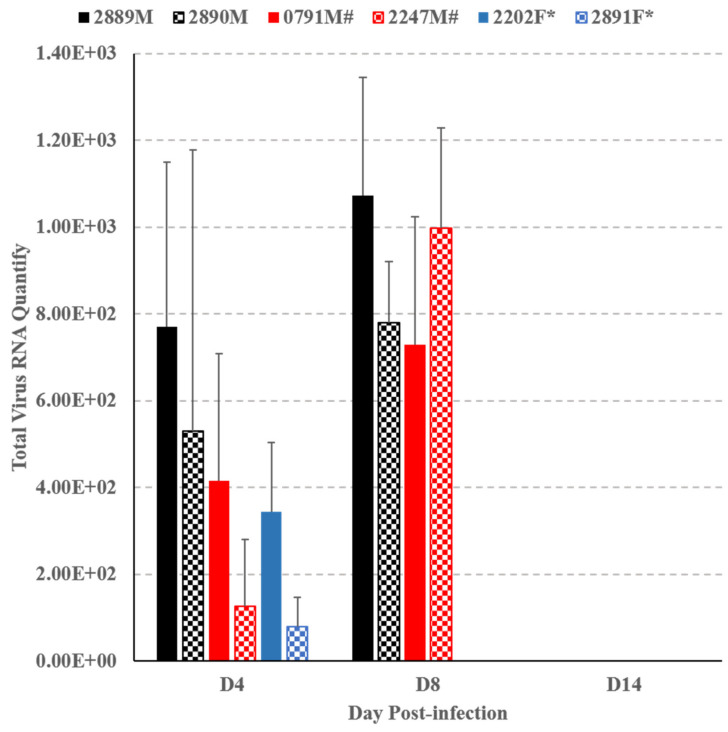
Total viral RNA quantity in the serum samples of monkeys collected on day 4, day 8 and day 14 postinfection was assayed by quantitative RT-PCR. Two of the monkeys were sacrificed on day 4 (2202F*, 2891F*) and two additional monkeys on day 8 (0791#, 2247#). No viral RNA was detected in the serum samples collected on day 14 of two remaining monkeys (2889M, 2890M).

**Table 1 viruses-13-00438-t001:** The number of nucleotide (NT), and corresponding amino acid (AA), mutations that occurred in each of the genomic segments of EV71:TLLα and EV71:TLLαP20 compared with their original parental wild-type.

Viral Gene Region/Protein		EV71:TLLα		EV71:TLLαP20	
		NT	AA	NT	AA
5′-UTR	“Cloverleaf”				
(1–747)	IRES	1		2	
P1	VP4	2	1	3	1
(748–3333)	VP2	2	2	2	2
	VP3	2	2	3	3
	VP1	2	2	4	3
P2	2A	2	2	4	3
(3334–5067)	2B	1		1	
	2C	2	1	2	1
P3	3A			1	1
(5068–7326)	3B				
	3C	1	1	1	1
	3D	3	1	3	1
3′-UTR					
(7327–7412)					
Total		18	12	26	16

**Table 2 viruses-13-00438-t002:** The number of nucleotide (NT), and corresponding amino acid (AA), mutations that occurred in each of the genomic segments of EV71:TLLβ, EV71:TLLβP20 and EV71:TLLβP40 compared with their original parental wild-type.

Viral Gene Region/Protein		EV71:TLLβ		EV71:TLLβP20		EV71:TLLβP40	
		NT	AA	NT	AA	NT	AA
5′-UTR	“Cloverleaf”	2		1		2	
(1–746)	IRES	2		2		3	
P1	VP4					1	
(747–3332)	VP2	4		4		4	
	VP3	2	2	2	2	2	2
	VP1	9	8	8	8	8	8
P2	2A	4	2	4	2	4	2
(3333–5066)	2B	1	1	1	1	1	1
	2C	2		3		3	
P3	3A	1	1	1	1	1	1
(5067–7325)	3B						
	3C	2	2	2	2	2	2
	3D	2	2	2	2	2	2
3′-UTR		1		1		1	
(7326–7411)							

Total		31	18	30	18	33	18

**Table 3 viruses-13-00438-t003:** The titre (unit in term of fold of dilution) of binding antibodies (IgM and IgG) from monkeys given intravenous EV71:TLLβP20. Titres were determined by indirect immunofluorescence using inhouse prepared EV-A71 subgenogroup B3 infected Vero cells as antigen.

Monkey	IgM titre		IgG titre	
	Day 14 PI	Day 30 PI	Day 14 PI	Day 30 PI
2889M	1:20	1:10	1:160	1:320
2890M	1:40	1:40	1:160	1:640

**Table 4 viruses-13-00438-t004:** The titre (unit in term of fold of dilution) of neutralizing antibodies against a number of subgenogroups of EV-A71 (A, B3, B4, B5, C1 and C5) present in the sera of two monkeys given intravenous EV71:TLLβP20. Titres were determined by microneutralization assay.

EV71 Genotype	Monkey			
	2889M		2890M	
	Day 14 PI	Day 30 PI	Day 14 PI	Day 30 PI
A (BrCr)	1:20	1:40	1:40	1:80
B3	1:160	1:640	1:160	1:640
B4	1:160	1:640	1:160	1:320
B5	1:80	1:320	1:160	1:320–640
C1	1:40	1:160	1:40	1:160
C5	1:20	1:40	1:20	1:40

## Data Availability

Not applicable.

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
