# Peer review of "Safety and Immunogenicity of a Stable, Cold-Adapted, Temperature-Sensitive/Conditional Lethal Enterovirus A71 in Monkey Study"

_viruses, 2021, doi:10.3390/v13030438_

Round 1
Reviewer 1 Report
The authors addressed most of the previous comments and suggestions and revised the manuscript accordingly.
Author Response
Thanks
Reviewer 2 Report
This manuscript describes the generation of cold-adapted EV71. The generated virus was tested in cell culture and monkeys for phenotype and genetic stability, immunogenicity and virulence. The authors concluded the EV71:TLLβ as stable in cell culture, and non-neurovirulent but immunogenic in monkeys.
My questions are listed below.
1. Monkey test is valuable to vaccine research but the lack of control has made it less informative. Although it has been indicated that the original strain of EV71:TLLβ was not tested in monkeys due to ethical and financial limitations, it still raises concerns. The parental strain of the EV71:TLLβ was isolated from a HFMD case, and it will be helpful to clarify whether this virus strain possess neurovirulence in monkeys.
Indeed, the generated EV71:TLLβ did not cause disease in infected monkeys (and barely replicate) based on the presented data, but it is unclear whether this was due to further attenuation by the cold-adaptation strategy or the non-neurovirulent phenotype of the parental strain. Although the authors did not claim the non-neurovirulent phenotype of the EV71:TLLβ was due to further attenuation by the cold adaptation but the descriptions in lines 79-81 and 578-579 may mislead the readers.
2. Serial passages in cell culture generates a diverse virus population, especially for enterovirus with high mutation rate. The authors identified several highly conserved dominant mutations that may be responsible for the cold adapted phenotype with Sanger sequencing. Have the authors generated a virus carrying these conserved mutations and whether this virus recapitulates the cold-adapted phenotype? And why the authors decided to proceed with a virus population but not a virus with defined genome composition as reverse genetics has been widely used in enterovirus studies.
3. The cold-adapted poliovirus has been reported by Sanders et al (authors’ reference #55). Similarly, determinants responsible for the cold-adapted PV1 were not validated and they only proposed their cold-adapted virus as a strain for inactivated vaccine.
Taken these together, the authors should provide in-depth discussion to justify the potential of this cold-adapted EV71 as live vaccine, or tone down to report the cold-adaptation as a strategy with potential to develop live EV71 vaccine.
4. Number of days to full CPE.
For the generation of cold-adapted virus, it is fine to harvest upon full CPE. However, to test growth characteristics as shown in figure 2, cells should be infected at the same m.o.i. for the same amount of time to determine titers of the generated viral progeny. Current figure 2b does not reflect the changes in replication ability of the passaged virus population, but only the virus titers when the infection of the cells was saturated.
5. Please add reference for the poliovirus 1 described in lines 506-508.
Author Response
Comments and Suggestions for Authors
This manuscript describes the generation of cold-adapted EV71. The generated virus was tested in cell culture and monkeys for phenotype and genetic stability, immunogenicity and virulence. The authors concluded the EV71:TLLβ as stable in cell culture, and non-neurovirulent but immunogenic in monkeys.
My questions are listed below.
- Monkey test is valuable to vaccine research but the lack of control has made it less informative. Although it has been indicated that the original strain of EV71:TLLβ was not tested in monkeys due to ethical and financial limitations, it still raises concerns. The parental strain of the EV71:TLLβ was isolated from a HFMD case, and it will be helpful to clarify whether this virus strain possess neurovirulence in monkeys.
Indeed, the generated EV71:TLLβ did not cause disease in infected monkeys (and barely replicate) based on the presented data, but it is unclear whether this was due to further attenuation by the cold-adaptation strategy or the non-neurovirulent phenotype of the parental strain. Although the authors did not claim the non-neurovirulent phenotype of the EV71:TLLβ was due to further attenuation by the cold adaptation but the descriptions in lines 79-81 and 578-579 may mislead the readers.
Our sincere apology for failure to demonstrate earlier whether the original parental wild-type strain is neurovirulence in this monkey study.
- Serial passages in cell culture generates a diverse virus population, especially for enterovirus with high mutation rate. The authors identified several highly conserved dominant mutations that may be responsible for the cold adapted phenotype with Sanger sequencing. Have the authors generated a virus carrying these conserved mutations and whether this virus recapitulates the cold-adapted phenotype? And why the authors decided to proceed with a virus population but not a virus with defined genome composition as reverse genetics has been widely used in enterovirus studies.
Thank you for your interest. We have not undertaken experimental study to determine the molecular determinants of this cold-adapted, temperature sensitive/conditional EV-A71. We have generated the required infectious cDNA of EV71:TLLβP20 to explore its use as enteroviral vector for human Enterovirus species A. Of course, with sufficient financial and man-power support, we are very keen and will undertake the experimental study.
I proceeded with a virus population but not a virus with a defined genome composition because my accumulated knowledge and experiences with human enteroviruses convinced me that the phenotypic and genetic characteristics of these enteroviruses are the outcome of a complex combinatorial interactions of various non-structural genes and non-coding 5’ and 3’ nucleotide sequences. In fact, as mentioned in the Material and Method, I started the initial screen of 10 EV-A71 strains and subsequently narrowed down to 3 for the cold-adaptation process. As stated in the Result, the 3 selected strains gave different phenotypic characteristics after 100 passages through the cold-adaptation processes. I acknowledge that I was lucky to have obtained the required virus strain, EV71:TLLβ. There is consideration of using the EV71:TLLαP20 for development of candidate inactivated EV-A71 vaccine.
- The cold-adapted poliovirus has been reported by Sanders et al (authors’ reference #55). Similarly, determinants responsible for the cold-adapted PV1 were not validated and they only proposed their cold-adapted virus as a strain for inactivated vaccine.
Taken these together, the authors should provide in-depth discussion to justify the potential of this cold-adapted EV71 as live vaccine, or tone down to report the cold-adaptation as a strategy with potential to develop live EV71 vaccine.
Thank you for your comment. We did mention and discuss the cold-adapted, temperature sensitive/conditional EV71:TLLβP20 as the potential live attenuated vaccine candidate.
- Number of days to full CPE.
For the generation of cold-adapted virus, it is fine to harvest upon full CPE. However, to test growth characteristics as shown in figure 2, cells should be infected at the same m.o.i. for the same amount of time to determine titers of the generated viral progeny. Current figure 2b does not reflect the changes in replication ability of the passaged virus population, but only the virus titers when the infection of the cells was saturated.
Thank you for your comment. We did use the same m.o.i (an inoculum of 10 m.o.i) in the infections as mentioned in the Material and Method.
- Please add reference for the poliovirus 1 described in lines 506-508.
Thank you. The previously mentioned reference was added accordingly. Page 15, Line 508.
Reviewer 3 Report
Authors have adressed all questions, thanks for their effort in impoving this manuscript.
Author Response
Thanks
Reviewer 4 Report
The study done by Chua and colleagues, described a promising Enterovirus A71 vaccine which is stable, cold-Adapted, temperature-sensitive/conditional lethal vaccine. The authors assessed the immunogenicity and safety of this candidate vaccines in monkeys, and all of the inoculated monkeys (except one) did not develop clinical symptoms of the disease/complications. On the other hand, the inoculated animals produced a good humoral immune response as shown by the increase the titre of IgM and IgG antibodies .
Overall the manuscript is well written, and it is important in this field since this potential vaccine candidate can be further developed to produce enteroviral vector and candidate oral live attenuated vaccine to protect children against EV-A71 infections and HFMD outbreak as mentioned by the authors in the conclusion.
I have minor comments
1) If the long term plan is to develop oral candidate vaccine, why the authors injected the animals by intravenous?, while not try to give the vaccine by oral gavage. Please write few lines about this in the discussion.
2)Minor text editing is needed in the introduction
a) page 1 line37: These include: please change to " These diseases include"
b) page 2 line 46-47, please add ref for sporadic outbreaks of neurovirulent EV-A71.
c) page 2 line 67" Neither is there any anti-viral drug 67
that has been approved for clinical treatment of severe infections and associated neurological complications. In terms of control and preventive strategies, the development of cost-effective vaccines against both CA16 and EV-A71, especially for use in developing countries, is of top public health priority". Please rephrase, it seems it a question not a statement.
d)page 2 line 76 : "This is based on vast past experiences" , please change to " This finding is based on vast past experiences"
Author Response
Comments and Suggestions for Authors
The study done by Chua and colleagues, described a promising Enterovirus A71 vaccine which is stable, cold-Adapted, temperature-sensitive/conditional lethal vaccine. The authors assessed the immunogenicity and safety of this candidate vaccines in monkeys, and all of the inoculated monkeys (except one) did not develop clinical symptoms of the disease/complications. On the other hand, the inoculated animals produced a good humoral immune response as shown by the increase the titre of IgM and IgG antibodies .
Overall the manuscript is well written, and it is important in this field since this potential vaccine candidate can be further developed to produce enteroviral vector and candidate oral live attenuated vaccine to protect children against EV-A71 infections and HFMD outbreak as mentioned by the authors in the conclusion.
I have minor comments
- If the long term plan is to develop oral candidate vaccine, why the authors injected the animals by intravenous?, while not try to give the vaccine by oral gavage. Please write few lines about this in the discussion.
Thank you for pointing out our short-coming. It is our oversight and assumption that it is a known “accepted standard” established by the Japanese virologists in National Institute of Infectious Diseases (NIID), Tokyo in assessing the neurovirulent of EV-A71. We have added a sentence in the Discussion section. Page 16, Lines 570-2.
2)Minor text editing is needed in the introduction.
- a) page 1 line37: These include: please change to " These diseases include"
Thank you. We have done the needful.
- b) page 2 line 46-47, please add ref for sporadic outbreaks of neurovirulent EV-A71.
Thank you for highlighting the point. We remove the word “sporadic” from the sentence to improve understanding. References for various epidemics and outbreaks were mentioned with respect to the specific events.
- c) page 2 line 67" Neither is there any anti-viral drug 67
that has been approved for clinical treatment of severe infections and associated neurological complications. In terms of control and preventive strategies, the development of cost-effective vaccines against both CA16 and EV-A71, especially for use in developing countries, is of top public health priority". Please rephrase, it seems it a question not a statement.
Thank you. We have done the needful.
d)page 2 line 76 : "This is based on vast past experiences" , please change to " This finding is based on vast past experiences"
Thank you. We have done the needful.
This manuscript is a resubmission of an earlier submission. The following is a list of the peer review reports and author responses from that submission.
Round 1
Reviewer 1 Report
In this paper, Chua and colleagues aimed to establish temperature-sensitive variants of enterovirus A71 (EV-A71) strains and validated in vitro and in vivo phenotypes of several cold adapted EV-A71 variants for the development of a live-attenuated EV-A71 vaccine. The methodological approaches, used in this study, to establish temperature-sensitive EV-A71 variants by a numerous number of serial passages in the cell culture at lower suboptimal temperatures would be reliable. The resultants cold-adapted EV-A71 variants (TLLα and TLLβ) exhibited a stable temperature-sensitive phenotype in Vero cells. As the authors mentioned, establishment of genetically stable live-attenuated EV-A71 strains is a unique and promising approach for further vaccine development. However, there are critical concerns with phenotypic characterization of the cold adapted EV-A71 variants, especially in a in vivo monkey model.
Specific comments
- As the authors mentioned (page 12), establishment of genetically stable live-attenuated EV-A71 strains is a unique and promising approach for further vaccine development just like Sabin OPV strains. However, most of the advantages of live-attenuated polio/enterovirus vaccines depend on oral immunization, but in this study, in vivo phenotypes (attenuation, immunogenicity, etc.) of the cold adapted EV-A71 variant were validated in a monkey model only by the iv inoculation. The authors should mention this point and discuss the limitation of the study very carefully.
- At least for the OPV strains, im injection is associated with provocation poliomyelitis via direct virus invasion from peripheral nerve to CNS so the possible risk of iv injection should be also considered.
- Although the number of monkeys have to be minimized from ethical points of view, I feel that several monkeys, inoculated with the original wild EV-A71 strain, should be arranged as an appropriate virulent control. Without such a control, the attenuation phenotype and immunogenicity of the cold adapted EV-A71 variant cannot be compared and validated in a monkey model.
- The genotype of the cold-adapted EV-A71 variants (TLLα and TLLβ) should be described in the text.
- In the result section, nt and amino acid substitutions of the cold-adapted EV-A71 variants from the wild strains should be described in more details. Are there any consensus substitutions between TLLα and TLLβvariants, etc.
- The quality of data presentation has to be improved (Fig. 3A and 3B).
Reviewer 2 Report
The authors passaged three EV-A71 strains at 28°C and generated three cold-adapted virus populations. Among them, EV-71:TLLβP20 was selected to be further tested in cell culture and monkey.
My questions are listed below.
- the article
- Cold-adapted poliovirus was previously reported as novel strains for inactivated vaccine in 2016 by Sanders et al. Since both EV-A71 and poliovirus belong to the same enterovirus genus, and the same approach was applied, the authors should consider to include this article in literature review.
- The authors emphasized the importance of vaccines for EV-A71 in introduction and discussion, described the generation and test of the cold-adapted virus population, but what is the conclusion of this study? How would this virus population benefit EV-A71 vaccine development? As a novel strain for live vaccine development or another safer strain for inactivated vaccine production.
- Also, the cold-adaptation strategy and why it was applied to developed the EV71 strain should also be discussed in the article.
- Stability of the cold-adapted temperature-sensitive phenotype.
- In the 4th paragraph of Section 3.1. The generated virus populations were passaged at 28°C for 20 passages and found maintained the phenotype. The virus populations were generated by passaging the parental strains with the same condition. Since the virus populations have been adapted to the temperature, additional passages are extended cold-adaptation or still the cold-selection process and provide little or no selection pressure to test stability of the temperature-sensitive phenotype. What is the rationale to use the same condition?
- The six passages under 37°C to test phenotype reversion. Passaging the cold-adapted virus population at higher temperature did provide selection pressure to test its stability, and the virus population evolved promptly. The result (Figure 2) showed that after only 2 passages, time for the virus to cause full CPE was shortened from four to two days, that is shorter than the three days for the original isolates. Also, the virus titers of passages at 28°C and 37°C were comparable (Figure 2b) since the 2nd This suggests the virus population evolved promptly to adapt to the higher temperature within 2 passages. When it took around 90 passages to become cold-adapted, only two passages at 37°C was enough for the virus population to regain the ability to replicate at 37°C and cause some CPE at 39.5°C. Thus, this phenotype doesn’t seem very stable as claimed.
- The authors concluded the virus population as genetic stable based on only one or two genetic changes from the 3rd passage. However, it is a diverse virus population generated by cold-adaptation, and tested at higher temperature. Sanger sequencing is not enough to determine genetic stability of the population as it only reveals the mutations that are fixed (or dominant) in the population. Changes in frequencies of the potential cold-adaptation determinants under selection pressures are better indicators for genetic stability and thus Next-Generation Sequencing should be considered.
- Monkey study.
- It is good that the antibodies induced by EV-71:TLLβP20 cross reacted with other EV-71 genotypes because different genotypic or subgenotypic EV-A71s emerged every few years and caused outbreaks. What is the genotype of the original EV-A71 strain? What are the clinical symptoms caused by this strain in human infection? The only information is that it was isolated from a stool specimen.
- Has the original strain been tested in monkey? If so, does it cause any observable disease in monkey or display neurovirulence?
- Two monkeys received booster and the antibody titers increases only 2-4-fold. Was this low increase in titers after booster expected?
Others:
- Not many genetic changes were revealed after the cold-adaptation process, have the authors tried to identify the determinants by generating mutant viruses?
- In figures 1 and 4, what is the unit of “Total virus RNA quantity”? RNA copy number? Was there a standard? And what is the error bar, range?
- Was experiment if figure 2b repeated or performed in replicates? Statistical analysis?
- Tables 1 and 2 are not very informative with only number of changes are provided. Nucleotide positions and/or amino acid positions can be added so that the genetic changes between passages are clearer.
- Only one sequence is available on GenBank. Maybe they were made available upon acceptance of the manuscript.
Reviewer 3 Report
Safety and Immunogenicity of a Stable, Cold-Adapted, Temperature sensitive/Conditional Lethal Enterovirus A71 in Monkey Study
This work presented here by Kaw Bing Chua et al are interesting for the development of new safe vaccine candidates against hand-foot-month disease. The authors have generated a stable, cold-adapted, temperature sensitive/conditional lethal EV-A71 through adaptative passaging of the isolated virus on Vero cells at low temperature (28°C). By temperature reversion study the authors showed phenotypic and genetic stability of the virus. They explain this stability by six unique mutations: 2 in non-coding regions and 4 in non-structural protein-coding genes. This result is very interesting, but I still have some questions:
- Among 6 monkeys used, two developed good humoral immune response on Day 14 and Day 30 post-inoculation, is this result statistically significant?
- Introduction, 3th paragraph: I guess that the authors want to say EV-A71?
- In the material and method section, the authors claim that histology and the immunohistochemistry were performed on CNS tissue specimens, but they don’t show any results (not figure nor supplementary data).
Which monoclonal antibody specific for the virus was used in Figure 3a?
- Results section:
3.1. Virus Phenotypic and Genotypic Characteristics in Cells
- From 1st to 4th paragraph, as the author do not show any data for these results, maybe it will be clearer and more interesting to include in figure 1 a graphical abstract for this section summarizing the results?
- 2+CPE: what does this mean? one figure presenting these data will be welcome.
- as there is no CPE noted with EV71-TLL even the author observed positive IFA, how the author explained that the positive IFA observed is not just the inoculum of the virus?
- Results for figure 1: the authors should explain and conclude their results/finding (table1 and2) not just present the data.
- Page 8, The number of nucleotide changes/reversions with respect to that of
EV71:TLLβP20, are shown in Table S1 (nb: I don’t get the supplementary table 1 and 2 in the PDF) :this result is very interesting I would like to see it in the figure.
3.2. Monkey Study
- Figure 3b: the authors should verify the quality of the Gel! The Gel2 was just copied and paste!
- page 11: can the authors include the table S2 in the figure 4?
- Table 3: what is the unit of IgG, IgM and neutralizing antibodies? Table 3a: please correct to IgG titer? The titer of neutralizing antibodies is presented as for example 1:20, 1:160…. What does this mean?
6- the authors showed that EV71:TLLβ contains 6 unique mutations that are highly conserved in all isolates. They show that some of these mutations are similar to those observed for Sabin vaccine poliovirus 1 which contribute to temperature-sensitive attenuation of the virus. This is very interesting and promising finding. Do the author plan to experience the role of the 6 unique mutations each and/or combined in direct generation by molecular biology of live attenuated vaccine candidate?